# A Quantum Information Theoretic Approach to Tractable Probabilistic Models

**Pedro Zuidberg Dos Martires**[1]

[1]Örebro University, Sweden

## Abstract

By recursively nesting sums and products, probabilistic circuits have emerged in recent years as an attractive class of generative models as they enjoy, for instance, polytime marginalization of random variables. In this work we study these machine learning models using the framework of quantum information theory, leading to the introduction of *positive unital circuits* (PUnCs), which generalize circuit evaluations over positive real-valued probabilities to circuit evaluations over positive semi-definite matrices. As a consequence, PUnCs strictly generalize probabilistic circuits as well as recently introduced circuit classes such as PSD circuits.

## 1 INTRODUCTION

Probabilistic circuits (PCs) [Darwiche, 2003, Poon and Domingos, 2011] belong to an unusual class of probabilistic models: they are highly expressive but at the same time also tractable. For instance, so-called decomposable probabilistic circuits [Darwiche, 2001a] encode probability distributions using nested sums and products over positive real-valued numbers and allow for the computation of marginals in time polynomial in the size of the circuit. Zhang et al. [2020] noted that it is exactly this restriction to positive values that limits the expressive efficiency (or succinctness) of PCs [Martens and Medabalimi, 2014, de Colnet and Mengel, 2021]. In particular, the positivity constraint on the set of elements that PCs operate on prevents them from modelling negative correlations between variables.

Circuits that are incapable of modelling negative correlations, i.e. circuits that can only combine probabilities in an additive fashion, are also called monotone circuits [Shpilka and Yehudayoff, 2010]. This restricted expressiveness can be combatted by the use of so-called *non-monotone* circuits,

where subtractions are allowed as a third operation (besides sums and products). Interestingly, Valiant [1979] showed that a mere single subtraction can render non-monotone circuits exponentially more expressive than monotone circuits – a result that has recently been refined for a subclass of decomposable circuits [Loconte et al., 2025b].

As shown by Harviainen et al. [2023] and Agarwal and Bläser [2024], non-monotone circuits do, however, introduce an important complication: if non-monotone circuits are not designed carefully, verifying whether a circuit encodes a valid probability distribution or not is an NP-hard problem. This does also render learning the parameters of a circuit practically infeasible.

Using the concept of *positive operator valued measures* from quantum information theory, which encode random events as positive semi-definite matrices, we are able to devise non-monotone circuits that nonetheless encode proper (normalized) probability distributions by construction. Our approach extends a line of recent works presented in the circuit literature [Sladek et al., 2023, Loconte et al., 2024, Wang and Van den Broeck, 2025, Loconte et al., 2025b]. However, our work is the first that establishes this deep connection between concepts in quantum information theory and tractable probabilistic models. Furthermore, the non-monotone circuits that we introduce generalize probabilistic circuits and PSD circuits [Sladek et al., 2023, Loconte et al., 2024, 2025b][1].

The remainder of the paper is structured as follows. We introduce in Section 2 the necessary concepts from quantum information theory. In Section 3 we then use these concepts to construct tractable probability distributions using positive operator circuits. In Section 4 we impose specific restrictions on the functional form of the computation units in PUnCs and show how these restrictions lead to circuit classes known in the literature, e.g. probabilistic circuits in Section 4.3.

---

[1]PSD circuits were later on rebranded as sum of compatible squares circuits [Loconte et al., 2025b]

In Section 5 we then drop the so-called property of *structured decomposability*, which has been imposed so far on all non-monotone circuit models. As such, we introduce the first circuit model that is non-monotone and only adheres to the weaker property of *decomposability*. We discuss related work in Section 6 and end the paper with concluding remarks in Section 7.

## 2 A PRIMER ON QUANTUM INFORMATION THEORY

A widely used and elegant framework to describe measurements of quantum systems is the so-called *positive operator-valued measure* (POVM) formalism. While POVMs have physical interpretations in terms of quantum information and quantum statistics, we will only be interested in their mathematical properties as we use them to show that circuits (defined in Section 3) form valid probability distributions. We refer the reader to [Nielsen and Chuang, 2001] for an in-depth exposition on the topic, as well as quantum computing and quantum information theory in general.

**Definition 2.1** (Positive Semidefinite). A $B \times B$ Hermitian matrix $H$ is called positive semi-definite (PSD) if and only if $\forall \mathbf{x} \in \mathbb{C}^B : \mathbf{x}^* H \mathbf{x} \geq 0$, where $\mathbf{x}^*$ denotes the conjugate transpose and $\mathbb{C}^B$ the $B$-dimensional space of complex numbers.

**Definition 2.2** (POVM [Nielsen and Chuang, 2001, Page 90]). A positive operator-valued measure is a set of PSD matrices $\{E(i)\}_{i=0}^{I-1}$ ($I$ being the number of possible measurement outcomes) that sum to the identity:

$$\sum_{i=0}^{I-1} E(i) = \mathbb{1}, \tag{1}$$

Before defining the probability of a specific $i$ occurring, we need the notion of a density matrix [von Neumann, 1927, Landau, 1927]:

**Definition 2.3** (Density Matrix [Nielsen and Chuang, 2001, Page 102]). A density matrix $\rho$ is a PSD matrix of trace one, i.e. $\mathrm{Tr}[\rho] = 1$.

**Definition 2.4** (Event Probability [Nielsen and Chuang, 2001, Page 102]). Let $\rho$ be a density matrix and let $i$ denote an event with $E(i)$ being the corresponding element from the POVM. The probability of the event $i$ happening, i.e. measuring the outcome $i$, is given by

$$p(i) = \mathrm{Tr}[\rho E(i)] \tag{2}$$

**Proposition 2.5.** *The expression in Equation 2 defines a valid probability distribution.*

*Proof.* While this is a well-known result we were not able to identify a concise proof in the literature. We therefore provide one in Appendix A.1. □

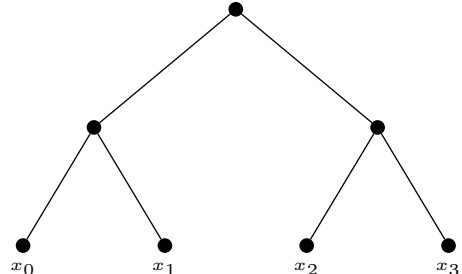

Figure 1: Partition circuit over four binary variables $x_i$ with $i \in \{0, 1, 2, 3\}$, which are given as inputs to the circuit. The internal nodes of the partition circuit correspond the computation units.

Given that the $E(i)$'s completely describe the event $i$ such that its event probability can be computed, they represent the quantum state of a system. This quantum state (represented by a matrix) lives in a certain Hilbert space. The changes that a quantum state can undergo are then described by so-called *quantum operations* acting on the Hilbert space. We can construct such operations using Kraus' theorem.

**Theorem 2.6** (Kraus' Theorem [Kraus, 1983]). *Let $\mathcal{H}$ and $\mathcal{G}$ be Hilbert spaces of dimension $N$ and $M$ respectively, and $\Phi$ be a quantum operation between $\mathcal{H}$ and $\mathcal{G}$. Then, there are matrices $\{K_j\}_{j=1}^{D}$ (with $D \leq NM$) mapping $\mathcal{H}$ to $\mathcal{G}$ such that for any state $E(i)$*

$$\Phi(E(i)) = \sum_{j=1}^{D} K_j E(i) K_j^* \tag{3}$$

*provided that $\sum_j K_j^* K_j \leq \mathbb{1}$ (in the Loewner order sense).*

*Proof.* See [Nielsen and Chuang, 2001, Chapter 8] □

The $K_j$ matrices are usually referred to as Kraus operators.

## 3 POSITIVE UNITAL CIRCUITS

A popular subclass of probabilistic circuits are so-called structured decomposable probabilistic circuits [Darwiche, 2011] that are also smooth [Darwiche, 2001b]. The advantage of this circuit subclass is that they can be implemented in a rather straightforward fashion on modern AI accelerators, as demonstrated by Peharz et al. [2019, 2020]. For the sake of exposition, we will limit ourselves in a first instance to such circuits that adhere to structured decomposability and will generalize to (non-structured) decomposable circuits in Section 5. For a detailed account on these different circuit properties we refer the reader to [Vergari et al., 2021].

Zuidberg Dos Martires [2024] introduced an abstraction for these smooth structured decomposable circuits in the form of partition trees. We further refine this by introducing the concept of a *partition circuit*. We give such a circuit in Figure 1. Note, the concept of a partition tree, and hence a partition circuit, is related to the concept of a variable

tree [Pipatsrisawat and Darwiche, 2008]. However, partition circuits emphasize an interpretation as computation graphs, unlike variable trees.

**Definition 3.1** (Partition Circuit). A partition circuit over a set of variables is a parametrized computation graph taking the form of a binary tree. The partition circuit consists of two kinds of computation units: *leaf* and *internal* units (including a single *root*). Units at the same distance from the root form a layer. Furthermore, let $\xi_k$ denote the root unit or an internal unit. The unit $\xi_k$ then receives its inputs from two units in the previous layer, which we denote by $\xi_{k_l}$ and $\xi_{k_r}$. Each computation unit is input to exactly one other unit, except the root unit, which is the input to no other unit.

### 3.1 POSITIVE OPERATOR CIRCUITS

Using the concept of partition circuits we construct positive operator circuits. Positive operator circuits can be thought of as generalizing circuit evaluations with probabilities to circuit evaluations with PSD matrices.

**Definition 3.2** (Positive Operator Circuit (Partition Circuit)). Let $\mathbf{x} = \{x_0, \ldots, x_{N-1}\}$ be a set of discrete variables. We define an operator circuit as a partition circuit whose computation units take the following functional form:

$$O_k(\mathbf{x}_k) = \begin{cases} E_{x_k} & \text{if } k \text{ is leaf} \\ \Phi_k\Big(O_{k_l}(\mathbf{x}_{k_l}) \otimes O_{k_r}(\mathbf{x}_{k_r})\Big) & \text{else,} \end{cases} \quad (4)$$

where the $E_{x_k}$'s are quantum state matrices, and where the $\Phi_k$ quantum operations.

Note that using the Kronecker product between $O_{k_l}(\mathbf{x}_{k_l})$ and $O_{k_r}(\mathbf{x}_{k_r})$ is a sensible choice as it describes the joint state of both subsystems.

**Proposition 3.3.** *Positive operator circuits are PSD.*

*Proof.* We know that all the leaves carry PSD matrices as they describe quantum states. Passing these on recursively to the quantum operations in the internal units retains the positive semi-definiteness as the Kronecker product between two PSD matrices is again PSD. □

### 3.2 CONSTRUCTING A PROBABILITY DISTRIBUTION

In Section 2 we saw that we can construct a probability distribution using a density matrix $\rho$ and a positive operator-valued measure, with the latter being a set of PSD matrices (cf. Definition 2.2) that sum to the unit matrix. Using a positive operator circuit $O(\mathbf{x})$ we indeed have a set of PSD matrices. Namely, one for each instantiation of the $\mathbf{x}$ variables. We now introduce *positive unital (operator) circuits* (PUnCs) for which also the summation to the unit matrix holds.

**Definition 3.4.** We call a quantum operation *unital* if

$$\Phi_k(\mathbb{1}_{k_l} \otimes \mathbb{1}_{k_r}) = \Phi_k(\mathbb{1}_{k_l k_r}) = \mathbb{1}_k, \quad (5)$$

where $\mathbb{1}_k$, $\mathbb{1}_{k_l}$, $\mathbb{1}_{k_l k_r}$, and $\mathbb{1}_{k_r}$ denote unit matrices of appropriate size,

**Proposition 3.5.** *Unital quantum operations are valid in the sense that the inequality $\sum_j K_j^* K_j \leq \mathbb{1}$ holds for all unital quantum operations.*

*Proof.* See Appendix B.1 □

**Definition 3.6.** We call a positive operator circuit *unital* if the quantum operations $\Phi_k$ are unital, and if the sets $\{E_{x_k}\}_{x_k \in \Omega(X_k)}$ form a POVM for each $X_k$.

**Proposition 3.7.** *Let $\mathbf{X}$ denote a set of random variables with sample space $\Omega(\mathbf{X})$. Then the set $\{O(\mathbf{x})\}_{\mathbf{x} \in \Omega(\mathbf{X})}$ of positive unital circuits forms a POVM.*

*Proof.* See Appendix B.2 □

**Theorem 3.8.** *Let $\rho$ be a density matrix and $O(\mathbf{x})$ a positive unital circuit. The function*

$$p_{\mathbf{X}}(\mathbf{x}) = \text{Tr}[O(\mathbf{x})\rho] \quad (6)$$

*is a proper probability distribution over the random variables $\mathbf{X}$ with sample space $\Omega(\mathbf{X})$.*

*Proof.* This follows from Propositions 2.5 and 3.7 □

One of the outstanding properties of probabilistic circuits is that they are tractable – in the sense that they allow for poly-time marginalization of random variables. Positive unital circuits retain this property.

**Proposition 3.9.** *Positive unital circuits allow for tractable marginalization.*

*Proof.* (Sketch) The proof is rather straightforward and hinges on the fact that the quantum operations in the internal units are assumed to be computable in polytime and on the fact that the marginalization of a random variable is performed by pushing the sum to the corresponding leaf in the partition circuit. Analogous to the proof of Proposition 3.7. □

## 4 SPECIAL CASES

We will now make certain structural assumptions on the matrices representing the quantum states and the functional form of the quantum operations $\Phi$. By doing so, we obtain the PSD circuits introduced by Sladek et al. [2023] and (structured decomposable) probabilistic circuits as described by Peharz et al. [2020] as special cases (Section 4.2 and Section 4.3 respectively).

## 4.1 HADAMARD PRODUCT UNITS

First, however, we note that our formulation of PUnCs already encompasses canonical polyadic tensor decompositions [Carroll and Chang, 1970] – a popular choice in the circuit literature [Shih et al., 2021, Loconte et al., 2025a] to merge partitions that uses the Hadamard product instead of the Kronecker product.

Specifically, we observe that the Hadamard product between two matrices $A$ and $B$ can be rewritten using a Kronecker product;

$$A \circ B = P(A \otimes B)P^*, \qquad (7)$$

where $P$ is the semi-unitary partial permutation matrix selecting a principal sub-matrix [Visick, 2000, Corollary 2]. This also means that a quantum operation involving a Hadamard product can be rewritten using a Kronecker product:

$$
\begin{aligned}
\Phi(A \circ B) &= \sum_i K_i (A \circ B) K_i^* \\
&= \sum_i K_i P (A \otimes B) P^* K_i^* \\
&= \sum_i K_i' (A \otimes B) K_i'^* = \Phi'(A \otimes B)
\end{aligned}
\qquad (8)
$$

Note that, for $\Phi'$ to be unital it suffices that $\sum_i K_i K_i^* = \mathbb{1}$ as $P$ is semi-unitary ($PP^* = \mathbb{1}$).

From the discussion above we conclude that we can safely limit the discussion to circuits with Kronecker products as circuits with Hadamard products follow as a special case.

## 4.2 PURE QUANTUM STATES

As the matrices that represent quantum states are PSD, we can decompose them as follows using the spectral theorem:

$$O = \sum_j V_j \otimes V_j^*, \qquad (9)$$

with the $V_j$'s denoting the eigenvectors. As a special case we then have so-called pure states. That is quantum states constructed with a single eigenvector:

$$O = V \otimes V^*, \qquad (10)$$

We will show now that by restricting PUnCs to performing operations on pure quantum states gives us the special case of PSD circuits as introduced by Sladek et al. [2023], which we define first using a partition circuit.

**Definition 4.1.** Let $\mathbf{x} = \{x_0, \ldots, x_{N-1}\}$ be a set of $N$ discrete variables. A PSD circuit is a partition circuit whose computation units take the following functional form:

$$
V_k(\mathbf{x}_k) =
\begin{cases}
U_k \times e_{x_k}, & \text{if } k \text{ leaf} \\
U_k \times (V_{k_l}(\mathbf{x}_k) \otimes V_{k_r}(\mathbf{x}_k)), & \text{else}
\end{cases}
\qquad (11)
$$

where the $U_k$'s are semi-unitary matrices. The probability $p_{\mathbf{X}}(\mathbf{x})$ is computed via

$$p_{\mathbf{X}}(\mathbf{x}) = V_{root}^*(\mathbf{x}) \times \rho \times V_{root}(\mathbf{x}), \qquad (12)$$

where $\rho$ is a density matrix.

Note that in the original formulation Sladek et al. [2023] used non-semi-unitary matrices. However, Loconte and Vergari [2024] have recently shown that there is no loss in expressiveness with such a restriction.

To show that PSD circuits are a special case of PUnCs we now impose the following restriction on the quantum operations $\Phi_k$:

$$\Phi_k(O_{k_l} \otimes O_{k_l}) = K_k (O_{k_l} \otimes O_{k_r}) K_k^* \qquad (13)$$

That is, we limit the quantum operation to having only a single pair of Kraus operators. For the quantum operation to be unital we need to have $K_k K_k^* = \mathbb{1}$. That is, $K_k$ has to be semi-unitary,

Furthermore, we make the following choice in the leaves:

$$E_{x_k} = K_k \left( e_{x_k} \otimes e_{x_k}^* \right) K_k^*, \qquad (14)$$

where the set $\{e_{x_k}\}_{x_k \in \Omega(X_k)}$ is a complete set of orthonormal basis vectors, and $K_k$ is again semi-unitary.

We can show that this choice for $E_{x_k}$ forms a POVM. Firstly, by observing that each $E_{x_k}$ is PSD. Secondly, by verifying the completeness of the set of operators:

$$
\begin{aligned}
\sum_{x_k \in \Omega(X_k)} E_{x_k} &= \sum_{x_k \in \Omega(X_k)} K_k \left( e_{x_k} \otimes e_{x_k}^* \right) K_k^* \\
&= K_k \left( \sum_{x_k \in \Omega(X_k)} e_{x_k} \otimes e_{x_k}^* \right) K_k^* \\
&= K_k \mathbb{1} K_k^* = \mathbb{1}
\end{aligned}
\qquad (15)
$$

**Definition 4.2.** We call a positive unital circuit pure if Equation 13 and Equation 14 hold.

**Proposition 4.3.** *For computation units of a pure positive unital circuit and a PSD circuit it holds that*

$$\forall k : O_k(\mathbf{x}_k) = V_k(\mathbf{x}_k) \otimes V_k^*(\mathbf{x}_k). \qquad (16)$$

*given that $U_k = K_k$*

*Proof.* See Appendix C.1 □

**Corollary 4.4.** *Pure positive unital circuits perform operations on pure quantum states exclusively.*

*Proof.* This follows immediately from Proposition 4.3. □

**Proposition 4.5.** *A PSD circuit and a pure PUnC encode the same probability distribution if $U_k = K_k$ for each unit $k$.*

*Proof.* See Appendix C.2 □

In this subsection we have shown that by making specific choices in the functional form of the leaves and the internal units of a positive unital circuit we recover the special case of PSD circuits and its variants [Loconte et al., 2024, 2025b]. Furthermore, our analysis also provides the rather satisfying interpretation of PSD circuits as quantum circuits acting on pure states exclusively. While this connection has already been pointed out informally by Wang and Van den Broeck [2025], we give a formal argument.

## 4.3 DIAGONAL POSITIVE UNITAL CIRCUITS

The oldest and most widely used class of tractable circuits fall into the model class of probabilistic circuits. These probabilistic circuits are well understood, and their properties have been mapped out comprehensively [Vergari et al., 2021]. We now show how we retrieve probabilistic circuits as a special case from PUnCs by imposing specific constraints on the functional form of the computation units of PUnCs. Specifically, by ensuring that the computations performed in PUnCs are closed over diagonal matrices. Before imposing diagonal closedness on PUnCs, we start by giving a definition of probabilistic circuits in terms of a partition circuits.

**Definition 4.6** (Probabilistic Circuit (Partition Circuit)). Let $\mathbf{x} = \{x_0, \ldots, x_{N-1}\}$ be a set of discrete variables. We define a probabilistic circuit as a partition circuit whose computation units take the following functional form:

$$P_k(\mathbf{x}_k) = \begin{cases} P_{x_k} & \text{if } k \text{ is leaf} \\ W_k \times \left( P_{k_l}(\mathbf{x}_{k_l}) \otimes P_{k_r}(\mathbf{x}_{k_r}) \right) & \text{else,} \end{cases}$$

where the $P_{x_k}$'s are real-valued positive vectors such that summing over $x_k$ gives a vector with exclusively ones as entries ($\sum_{x_k} P_{x_k} = [1, \ldots, 1]^T$), and where the $W_k$'s are row-normalized matrices with positive entries only, i.e. $\forall k, i : \sum_j W_{kij} = 1$, where the $i$ and $j$ indices index the matrix. Furthermore, the dimensions of the $W_k$'s, $P_{x_k}$'s and $P_k(\mathbf{x}_k)$ are such that they match the matrix-vector products in the computation units.

**Proposition 4.7.** *Every entry of a vector $P_k(\mathbf{x}_k)$ forms a valid probability distribution.*

*Proof.* The proof is included for the sake of completeness in Appendix C.3 and follows a similar structure to those found in the literature, e.g. [Peharz et al., 2015]. □

Next, we define *diagonal* PUnCs. For this, consider the following functional form of a quantum operation:

$$\Phi(O) = \sum_j J_j D_j O D_j^* J_j^*, \qquad (17)$$

where the $D_j$'s are diagonal matrices such that $\mathrm{Tr}[D_j D_j^*] = 1$. The $J_j$ are sparse matrices that are zero

everywhere but in the $j$-th row where all their entries are 1. For instance, if we assume that $J_2$ is a 3 by 3 matrix it takes the following form: $J_2 = \begin{pmatrix} 0 & 0 & 0 \\ 1 & 1 & 1 \\ 0 & 0 & 0 \end{pmatrix}$. Together, $J_j$ and $D_j$ represent the Kraus operators $K_j = J_j D_j$.

Given that $O$ is PSD it is obvious that also $\Phi(O)$ in Equation 17 is PSD: the individual term of the sum on the right-hand side are all PSD and the sum of PSD matrices is again a PSD matrix. For showing that the quantum operation is also unital, we rewrite the expression as follows:

$$\Phi(\mathbb{1}) = \sum_j J_j \text{diagmat}(w_j) J_j^*, \qquad (18)$$

where $w_j$ is a vector whose entries correspond to the diagonal elements of the matrix $D_j D_j^*$. Simply carrying out the matrix products results in:

$$\Phi(\mathbb{1}) = \sum_j H_j \sum_i w_{ji}, \qquad (19)$$

where $H_j$ is a square matrix that is zero everywhere but on the $j$-th entry of the diagonal where it is 1. From the condition that $\mathrm{Tr}[D_j D_j^*] = 1$ it immediately follows that $\sum_i w_{ji} = 1$. This leaves us with:

$$\Phi(\mathbb{1}) = \sum_j H_j = \mathbb{1}. \qquad (20)$$

Furthermore, in the leaves we pick

$$E_{x_k} = \Delta_{x_k} \Delta_{x_k}^*, \qquad (21)$$

such that the $\Delta_{x_k}$'s are diagonal matrices and such that $\sum_{x_k} \Delta_{x_k} \Delta_{x_k}^* = \mathbb{1}$.

**Definition 4.8.** We call a positive unital circuit diagonal if Equation 17 and Equation 21 hold.

**Proposition 4.9.** *All operators $O_k(\mathbf{x}_k)$ in a diagonal PUnC can be represented as diagonal matrices.*

*Proof.* See Appendix C.4 □

**Proposition 4.10.** *Probabilistic circuits and diagonal PUnCs are isomorphic.*

*Proof.* See Appendix C.5 □

This last proposition tells us that every (structured decomposable) probabilistic circuits can be represented as a diagonal PUnC (and vice versa).

## 4.4 BLOCK-DIAGONAL PUNCS

In order to combat theoretical limitations of pure and diagonal PUnCs, Loconte et al. [2025b] introduced a circuit class dubbed *μSOCS*. In Appendix D we show how this circuit class can be represented with PUnCs over block-diagonal matrices. Interestingly, this allows us to discuss the concept of noise in quantum information theory.

# 5 DECOMPOSABLE PUNCS

Using the concept of partition circuits we were able to define the different circuits in terms of matrix-matrix or matrix-vector multiplications. This was, however, only possible because the circuits we have studied so far obey the property of *structured decomposability* [Pipatsrisawat and Darwiche, 2008, Darwiche, 2011]. We will now study circuits adhering to the weaker property of (non-structured) decomposability. To this end, we first define probabilistic circuits in the usual way using nested sum and product units [Vergari et al., 2021] (and not partition circuits).

**Definition 5.1** (Probabilistic Circuit). Let $\mathbf{x} = \{x_0, \ldots, x_{N-1}\}$ be a set of discrete variables. A probabilistic circuit is a computation graph consisting of three kinds of computational units: *leaf*, *product*, and *sum*. Each product or sum unit receives inputs from a set of input units denoted by in($k$). Each unit $k$ encodes a function $p_k(\mathbf{x}_k)$ with $\mathbf{x}_k \subseteq \mathbf{x}$ as follows:

$$p_k(\mathbf{x}_k) = \begin{cases} f_k(x_k) & \text{if } k \text{ leaf unit} \\ p_{k_l}(\mathbf{x}_{k_l})p_{k_r}(\mathbf{x}_{k_r}) & \text{if } k \text{ product unit} \\ \sum_{j \in \text{in}(k)} w_{kj}p_j(\mathbf{x}_j) & \text{if } k \text{ sum unit} \end{cases}$$

where $f_{x_k}$ denotes a parametrized function such that $\sum_{x_k} f_{x_k} = 1$ and where $\forall k : \sum_{j \in \text{in}(k)} w_{kj} = 1$.

**Proposition 5.2.** *Let $\mathbf{X}_k$ be a set of random variables with sample space $\Omega(\mathbf{X}_k)$ equal to the possible values for $\mathbf{x}_k$. A probabilistic circuit $p(\mathbf{x}_k)$ then defines a proper probability distribution as $p_{\mathbf{X}_k}(\mathbf{x}_k) = p_k(\mathbf{x}_k)$.*

*Proof.* We need to show that $\forall \mathbf{x}_k \in \Omega(\mathbf{X}_k) : p_{\mathbf{X}_k}(\mathbf{x}_k) \geq 0$ and that $\sum_{\mathbf{x}_k \in \Omega(\mathbf{X}_k)} p_{\mathbf{X}_k}(\mathbf{x}_k) = 1$. Both of which are trivial. $\square$

The sums and products that we write explicitly in Definition 5.1 are also implicitly present in Definition 4.6, namely within the matrix-vector product of the internal computation units of the underlying partition circuit.

## 5.1 A PRIMER ON DECOMPOSABILITY

We can now define the concept of (non-structured) decomposability using the concept of a scope function.

**Definition 5.3** (Scope). The scope of a unit $k$, denoted by $\phi(k)$, is the set of random variables $\mathbf{X}_k$ for which the function $p_k(\cdot)$ encodes a probability distribution.

**Definition 5.4** (Decomposability). A circuit is decomposable if the inputs of every product unit $k$ encode distributions over disjoint sets of random variables: $\phi(k_l) \cap \phi(k_r) = \emptyset$ with $\{k_l, k_r\} = \text{in}(k)$.

It is precisely this decomposability property that enables tractable (any-order) marginalization in probabilistic circuits, as well as positive operator circuits and relaxing the

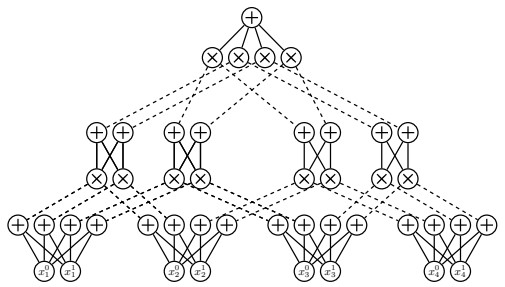

Figure 2: Graphical representation of a (non-structured) decomposable probabilistic circuit. We have four binary random variables in the leaves at the bottom. These are first passed individually through a set of sum units. In the second layer of the circuit we then have for blocks of computation units $\kappa_1$, $\kappa_2$, $\kappa_3$, and $\kappa_4$ (from lest to right). Each with their own scope ($\phi(\kappa_1) = \{X_1, X_2\}$, $\phi(\kappa_2) = \{X_1, X_3\}$, $\phi(\kappa_3) = \{X_2, X_4\}$, and $\phi(\kappa_4) = \{X_3, X_4\}$). At the root we have a block of computations $\kappa_{root}$ with $\phi(\kappa_{root}) = \{X_1, X_2, X_3, X_4\}$. Note that we slightly abuse notation and applied the scope function on sets of computation units instead of single computations units.

property of decomposability leads necessarily to a decrease in tractability [Choi et al., 2020, Vergari et al., 2021, Zuidberg Dos Martires, 2024]. Usually the smoothness property is also assumed to hold

**Definition 5.5** (Smoothness). A circuit is smooth if for every sum unit $k$ its inputs encode distributions over the same random variables: $\forall j_1, j_2 \in \text{in}(k)$ it holds that $\phi(j_1) = \phi(j_2)$.

Note that for circuits constructed using a partition tree, the decomposability and smoothness properties hold by construction. We give a graphical representation of a decomposable circuit in Figure 2.

**Definition 5.6** (Structured Decomposability). A circuit is structured decomposable if the circuit is decomposable and product units with identical scope decompose identically: let $k$ and $j$ be two product nodes. If we have that $\phi(k_l) = \phi(j_l)$ and $\phi(k_r) = \phi(j_r)$ for every pair of product nodes where $\phi(k) = \phi(j)$, then we call a circuit structured decomposable.

For partition circuits, we have again that the property of structured decomposability is respected by construction. We give a structured decomposable circuit in Figure 3.

We can now compare the two circuits in Figure 2 and Figure 3. Specifically, we point to the computational blocks at the root of the circuits, where we have for each circuit four product units and a single sum unit. In both circuits the scope of all the product units (and the sum unit) is the same:

$$\phi(p_k^{NSD}) = X_1, X_2, X_3, X_4 \tag{22}$$
$$\phi(p_k^{SD}) = X_1, X_2, X_3, X_4, \tag{23}$$

where $NSD$ stands for non-structured decomposable and $SD$ for structured decomposable. The index $k \in \{1, 2, 3, 4\}$ specifies in both cases the individual product units at the root (going from left to right). For the structured decomposable circuit in Figure 3 we have that the scope arises from the same union of random variables for all four units: $\phi(p_k^{SD}) = \{X_1, X_2\} \cup \{X_3, X_4\}$, for $k \in \{1, 2, 3, 4\}$.

The situation for the non-structured decomposable circuit presents itself differently. Here we have $\phi(p_k^{SD}) = \{X_1, X_2\} \cup \{X_3, X_4\}$, for $k \in \{2, 3\}$ and $\phi(p_k^{NSD}) = \{X_1, X_3\} \cup \{X_2, X_4\}$, for $k \in \{1, 4\}$ – showing that not all product units with the same scope decompose their respective scopes in the same fashion.

Pipatsrisawat and Darwiche [2008] showed that dropping the requirement that the product nodes with the same scope decompose in the same way leads to exponential gains in expressiveness. In the next subsection we show how we can construct non-structured decomposable non-monotone circuits. This is in contrast to all the non-monotone circuits discussed in the previous sections and the non-monotone circuits discussed in the literature [Sladek et al., 2023, Loconte et al., 2025b, 2024, Wang and Van den Broeck, 2025].

## 5.2 DROPPING STRUCTURED DECOMPOSABILITY

**Definition 5.7** (Positive Unital Circuit). Let $\mathbf{x} = \{x_0, \ldots, x_{N-1}\}$ be a set of discrete variables. A positive unital circuit is a computation graph consisting of three kinds of computational units: *leaf*, *product*, and *sum*. Each product or sum unit receives inputs from a set of input units denoted by $\mathrm{in}(k)$. Each unit $k$ encodes a function $o_k(\mathbf{x}_k)$ with $\mathbf{x}_k \subseteq \mathbf{x}$ as follows:

$$o_k(\mathbf{x}_k) = \begin{cases} e_{x_k} & \text{if } k \text{ leaf unit} \\ o_{k_l}(\mathbf{x}_{k_l}) \otimes o_{k_r}(\mathbf{x}_{k_r}) & \text{if } k \text{ product unit} \\ \sum_{j \in \mathrm{in}(k)} w_{kj} \Phi_{kj}(o_j(\mathbf{x}_j)) & \text{if } k \text{ sum unit} \end{cases}$$

where $e_{x_k}$ denotes an element of a POVM. The $\Phi_{kj}$ are unital quantum operations and the $w_{kj}$ are positive real-valued scalars and obey $\forall k : \sum_j w_{kj} = 1$.

Note how the sum units form convex combinations of quantum operation, i.e. quantum mixtures. From now on we will denote (non-structured) decomposable PUnCs by D-PUnCs (cf. Definition 5.7) and structured decomposable PUnCs by SD-PUnCs (cf. Section 3).

**Theorem 5.8.** *Let $\mathbf{X}_k$ be a set of random variables with sample space $\Omega(\mathbf{X}_k)$ equal to the possible values for $\mathbf{x}_k$. A D-PUnC $o(\mathbf{x}_k)$ and a density matrix $\rho$ then define a proper probability distribution as $p_{\mathbf{X}}(\mathbf{x}) = \mathrm{Tr}[o(\mathbf{x})\rho]$.*

*Proof.* See Appendix E.1 □

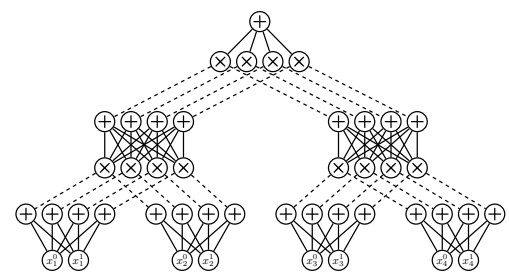

Figure 3: Graphical representation of a structured decomposable probabilistic circuit. We have four binary random variables in the leaves at the bottom. These are first passed individually through a set of sum units. In the second layer of the circuit we then have two blocks of computation units $\kappa_1$, $\kappa_2$ with scopes ($\phi(\kappa_1) = \{X_1, X_2\}$ and $\phi(\kappa_2) = \{X_3, X_4\}$). At the root we have a block of computations $\kappa_{root}$ with $\phi(\kappa_{root}) = \{X_1, X_2, X_3, X_4\}$. One can easily see how such a circuit maps on to a partition circuits by associating each block of computation units to a unit in a partition circuit, cf. Figure 1.

**Proposition 5.9.** *SD-PUnCs are a proper subset of D-PUnCs.*

*Proof.* See Appendix E.2 □

**Proposition 5.10.** *(Non-structured) decomposable probabilistic circuits are a proper subset of D-PUnCs.*

*Proof.* This follows trivially from the observation that restricting the circuit elements $o_k$ to $1 \times 1$ dimensional PSD matrices we are left with positive real-valued scalars. In this case the Kronecker product becomes the usual product over the reals and quantum operations simplify to identity operations, and Definition 5.7 is equivalent to that of a probabilistic circuit (Definition 5.1). □

**Proposition 5.11.** *D-PUnCs allow for tractable marginalization.*

*Proof.* The proof follows a similar rationale as the one for Proposition 3.9 □

To the best of our knowledge, we present with D-PUnCs the first non-monotone tractable circuit class that encodes (together with a denisty matrix) a positive function and that does not adhere to structured decomposability but only (non-structured) decomposability, regardless of the input. This means that similarly to how non-monotone structured decomposable circuits, e.g. SD-PUnCs and the works of [Sladek et al., 2023, Loconte et al., 2025b, Wang and Van den Broeck, 2025], generalize the language of (weighted) *SDNNFs* [Pipatsrisawat and Darwiche, 2008], D-PUnCs generalize the strictly larger language of *DNNFs* [Darwiche, 2001a].

# 6 RELATED WORK

PUnCs extend recent advancements in the probabilistic circuit literature (PSD circuits [Sladek et al., 2023], SOCS [Loconte et al., 2025b], and inception circuits [Wang and Van den Broeck, 2025]), which in turn extend the traditionally monotone circuits to the non-monotone setting.[2]. The main differentiator of PUnCs with regard to these works is that we can use decomposability instead of structured decomposability.

Furthermore, positive unital circuits provide also a different perspective on constructing non-monotone circuits. While the methods described in [Sladek et al., 2023, Loconte et al., 2024, 2025b, Wang and Van den Broeck, 2025] regard such circuits as sum of (nested) squares, we interpret them as probabilistic events described by positive semi-definite matrices that are combined within a circuit using unit preserving quantum operations. As such, we also establish a strong link between quantum information theory and the circuit literature.

As already pointed out by Loconte et al. [2024, 2025b,a] structured decomposable circuits share many aspects with *tensor networks* [Orús, 2014, White, 1992] – a class of statistical models developed in the condensed matter physics community, which have in recent years also been applied to supervised and unsupervised machine learning [Cheng et al., 2019, Han et al., 2018, Stoudenmire and Schwab, 2016]. In this regard, and given that tensor networks originate in the physics community, it is rather surprising that tensor networks have so far not been formulated using POVMs and quantum information theory.

First results on the expressive power of tensor networks and by extension of non-monotone circuits, were presented in the tensor network literature in the context of tensor decompositions [Glasser et al., 2019] and complex-valued hidden Markov models [Gao et al., 2022]. Recently, the works of Loconte et al. [2024, 2025b] and Wang and Van den Broeck [2025] have studied the relationship of different circuit classes more carefully, as well. Additionally, Loconte et al. [2025b] pointed out links between tensor networks and the circuit literature and were able to generalize earlier results from the tensor network literature by Glasser et al. [2019]. As the circuits studied by Loconte et al. [2025b] and Wang and Van den Broeck [2025] are special cases of D-PUnCs their results do also partially apply to D-PUnCs.

Lastly, we point out theoretical results in the statistical relational AI literature. Specifically, Buchman and Poole [2017b], and Kuzelka [2020] noted that using only real-valued parametrizations (including negatives [Buchman and Poole, 2017a]), does not allow for fully expressive models.

---

[2]We refer the reader to [Loconte et al., 2025b, Section 5] for a discussion on the relationship between inception circuits and SOCS.

# 7 CONCLUSIONS & FUTURE WORK

Based on first principles from quantum information theory, we constructed positive unital circuits – a novel class of probabilistic tractable models (Section 3). In a first instance, we then showed how structured decomposable PUnCs encompass all existing non-monotone circuit classes by enforcing certain constraints on the functional form of the quantum operations (Section 4). Then we continued in Section 5 with showing how the formalism of positive unital circuits is effortlessly relaxed to (non-structured) decomposable PUnCs– thereby creating a new circuit class.

In future work we would like to investigate in detail the expressive power of D-PUnCs compared to SD-PUnCs and (non-structured) decomposable probabilistic circuits. Specifically, we make the following conjecture.

**Conjecture 7.1.** *There is an exponential separation in expressive efficiency between D-PUnCs and SD-PUnCs.*

Effectively, this would expand the research initiated by Loconte et al. [2024] for mapping out the relationships between the different structured decomposable circuits. Ideally, one would like to create an analogue to Darwiche and Marquis' knowledge compilation map [Darwiche and Marquis, 2002] but for positive operator circuits and relate it to the framework of algebraic model counting [Kimmig et al., 2017]. This would also allow for establishing possible separations between sum-of-squares PCs [Loconte et al., 2024], product of monotonic by SOCS [Loconte et al., 2024, Definition 5], and inception networks [Wang and Van den Broeck, 2025]. A first discussion on these issues can also be fund in [Wang and Van den Broeck, 2025, Appendix A].

A more audacious goal would then be to investigate whether PUnCs can be run efficiently on quantum computers and whether there are speed-ups to be had. We can formulate this question more directly as a conjecture,

**Conjecture 7.2.** *There are circuit-query pairs that are tractable on a quantum computer but not on a classical computer.*

We stipulate that a positive answer to this question would involve Fourier transforms as they are the key mechanism underlying Shor's algorithm [Shor, 1994] leading to exponential quantum speed-up. Note that this question seems also related to the work of Riguzzi [2024]. Although the question in that work is on performing a computationally hard weighted model count using Grover's algorithm [Grover, 1996], which does not provide an exponential speed-up.

Finally, a more practical research avenue consists of finding parametrizations of PUnCs that sidestep the expensive dense matrix-matrix multiplications and replace them with structured sparse matrices. Monarch matrices seem to hold promise [Dao et al., 2022, Zhang et al., 2025].

## Acknowledgements

The work was supported by the Wallenberg AI, Autonomous Systems and Software Program (WASP) funded by the Knut and Alice Wallenberg Foundation.

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

# A Quantum Information Theoretic Approach to Tractable Probabilistic Models Appendix

**Pedro Zuidberg Dos Martires**[1]

[1]Örebro University, Sweden

## A  PROOFS FOR SECTION 2

### A.1  PROOF OF PROPOSITION 2.5

**Proposition 2.5.** *The expression in Equation 2 defines a valid probability distribution.*

*Proof.* While this is a well-known result we were not able to identify a concise proof in the literature and provide therefore, for the sake of completeness, one here. To this end, we first show that $p(i) \geq 1$, for each $i$.

$$p(i) = \mathrm{Tr}[E(i)\rho] = \mathrm{Tr}[D(i)D^*(i)\rho] \qquad (24)$$
$$= \mathrm{Tr}[D(i)^*\rho D(i)].$$

Here we used the fact that $E(i)$ is PSD and factorized it into the product $D(i)D^*(i)$. Then we used the fact that the trace is invariant under cyclical shifts. Clearly, the matrix $D(i)^*\rho D(i)$ is PSD as we have for every vector $x$:

$$x^* D(i)^* \rho D(i) x = y^* \rho y \geq 0 \quad \text{with } y = Dx. \qquad (25)$$

As the trace of a PSD matrix is positive we have that $p(i)$ is a positive real number, i.e. $p(i) \geq 0$ for every $i$.

Secondly, we show that $p(i)$ is normalized:

$$\sum_{i=1}^{N} p(i) = \sum_{i=1}^{N} \mathrm{Tr}[E(i)\rho] = \mathrm{Tr}\left[\sum_{i=1}^{N} E(i)\rho\right] = \mathrm{Tr}[\rho], \qquad (26)$$

where we used Equation 1. Exploiting the fact that the trace of a density matrix is 1 gives us indeed $\sum_{i=1}^{N} p(i) = 1$, and we can conclude that $p(i)$ is a valid probability distribution. □

## B  PROOFS FOR SECTION 3

### B.1  PROOF OF PROPOSITION 3.5

**Proposition 3.5.** *Unital quantum operations are valid in the sense that the inequality $\sum_j K_j^* K_j \leq \mathbb{1}$ holds for all unital quantum operations.*

*Proof.* The (sufficient and necessary) condition that $\sum_j K_j^* K_j \leq \mathbb{1}$ stems from fact that we wish to bound the probability of a state $\Phi(E(i))$ to be less than or equal to 1, cf. [Nielsen and Chuang, 2001, Proof of Theorem 8.1]. That is, we wish to have:

$$\mathrm{Tr}[\Phi(E(i))\rho] \leq 1. \qquad (27)$$

We will now show that for unital quantum operations this holds by construction and that the condition $\sum_j K_j^* K_j$ is implied. We start with the probability for the state $\sum_i E(i) = \mathbb{1}$:

$$\text{Tr}[\Phi(\mathbb{1})\rho] = \text{Tr}\left[\sum_j K_j \mathbb{1} K_j^* \rho\right] = \text{Tr}\left[\sum_j K_j K_j^* \rho\right] = \text{Tr}[\rho] = 1 \tag{28}$$

Alternatively, we write this also as:

$$\text{Tr}[\Phi(\mathbb{1})\rho] = \text{Tr}\left[\sum_j K_j \mathbb{1} K_j^* \rho\right] = \sum_j \text{Tr}\left[K_j \mathbb{1} K_j^* \rho\right] = \sum_j \text{Tr}\left[\gamma^* K_j \mathbb{1} K_j^* \gamma\right] \tag{29}$$

with $\rho = \gamma\gamma^*$.

Similarly, we also write for the probability of the arbitrary state $\sigma$ denoting the sum over any subset of the POVM $\{E(i)\}_{i=1}^N$:

$$\text{Tr}[\Phi(\sigma)\rho] = \sum_j \text{Tr}\left[\gamma^* K_j \sigma K_j^* \gamma\right]. \tag{30}$$

We now need that $\text{Tr}[\Phi(\sigma)\rho] \leq 1$, which is equivalent to:

$$1 - \text{Tr}[\Phi(\sigma)\rho] \geq 0 \Leftrightarrow \text{Tr}[\Phi(\mathbb{1})\rho] - \text{Tr}[\Phi(\sigma)\rho] \geq 0 \tag{31}$$

$$\Leftrightarrow \sum_j \text{Tr}\left[\gamma^* K_j \mathbb{1} K_j^* \gamma\right] - \sum_j \text{Tr}\left[\gamma^* K_j \sigma K_j^* \gamma\right] \geq 0 \tag{32}$$

$$\Leftrightarrow \sum_j \text{Tr}\left[\gamma^* K_j \left(\mathbb{1} - \sigma\right) K_j^* \gamma\right] \geq 0 \tag{33}$$

The last line only holds if $\mathbb{1} - \sigma$ is PSD, which is indeed the case as $\sigma$ only sums over a subset of the POVM. Subtracting the sum of this subset from $\mathbb{1}$ leaves us with a sum over the remaining elements of the POVM. As this is a sum over PSD matrices the sum over the remaining elements is again PSD. This concludes the proof as we have shown that Equation 27 is satisfied by construction. □

## B.2 PROOF OF PROPOSITION 3.7

**Proposition 3.7.** *Let* $\mathbf{X}$ *denote a set of random variables with sample space* $\Omega(\mathbf{X})$. *Then the set* $\{O(\mathbf{x})\}_{\mathbf{x} \in \Omega(\mathbf{X})}$ *of positive unital circuits forms a POVM.*

*Proof.* Given that positive unital circuits are by definition positive operator circuits we already have that:

$$\forall \mathbf{x} \in \Omega(\mathbf{X}) : O(\mathbf{x}) \text{ is PSD.} \tag{34}$$

Next we show that $\sum_{\mathbf{x} \in \Omega(\mathbf{X})} O(\mathbf{x}) = \mathbb{1}$. Here we observe that in the computation units we have:

$$\sum_{\mathbf{x}_k \in \Omega(\mathbf{X}_k)} O_k(\mathbf{x}_k) = \sum_{\mathbf{x}_{k_l}} \sum_{\mathbf{x}_{k_r}} \Phi(O_k(\mathbf{x}_{k_l}) \otimes O_k(\mathbf{x}_{k_r})) \tag{35}$$

$$= \Phi\left(\sum_{\mathbf{x}_{k_l}} O_k(\mathbf{x}_{k_l}) \otimes \sum_{\mathbf{x}_{k_r}} O_k(\mathbf{x}_{k_r})\right)$$

This lets us push down the summation of a specific variable to the corresponding leaf where the variable is given as input, where we then have:

$$\sum_{\mathbf{x}_k \in \Omega(\mathbf{X}_k)} O_k(\mathbf{x}_k) = \sum_{x_k \in \Omega(X_k)} E_{x_k} = \mathbb{1}_{\mathbb{k}} \tag{36}$$

We now exploit that the completely positive maps in a positive unital circuit are unital, which gives us indeed $\sum_{\mathbf{x} \in \Omega(\mathbf{X})} O(\mathbf{x}) = \mathbb{1}$. □

# C PROOFS FOR SECTION 4

## C.1 PROOF OF PROPOSITION 4.3

**Proposition 4.3.** *For computation units of a pure positive unital circuit and a PSD circuit it holds that*

$$\forall k : O_k(\mathbf{x}_k) = V_k(\mathbf{x}_k) \otimes V_k^*(\mathbf{x}_k). \tag{16}$$

*given that $U_k = K_k$*

*Proof.* We start the proof at the leaf where we have

$$O_k(x_k) = U_k \left( e_{x_k} \otimes e_{x_k}^* \right) U_k^* = (U_k e_{x_k}) \otimes \left( e_{x_k}^* U_k^* \right) \tag{37}$$

and Equation 16 holds almost by definition. In the computation units into which the leaves feed, we then have

$$\begin{aligned}
O_k &= U_k \left( O_{k_l} \otimes O_{k_r} \right) U_k^* \\
&= U_k \left( (V_{k_l} \otimes V_{k_l}^*) \otimes (V_{k_r} \otimes V_{k_r}^*) \right) U_k^* \\
&= \left( U_k \left( V_{k_l} \otimes V_{k_r} \right) \right) \otimes \left( \left( V_{k_l}^* \otimes V_{k_r}^* \right) U_k^* \right) \\
&= v_k \otimes v_k^*.
\end{aligned} \tag{38}$$

where we omitted the explicit dependencies on the variables $x_k$, $x_{k_l}$, and $x_{k_r}$. Repeating this argument recursively until the root of the circuit concludes the proof. $\square$

## C.2 PROOF OF PROPOSITION 4.5

**Proposition 4.5.** *A PSD circuit and a pure PUnC encode the same probability distribution if $U_k = K_k$ for each unit $k$.*

*Proof.* The proof starts by simply plugging in the vector representation of $O_{\text{root}}$ (obtained in the proof of Proposition 4.3) into the expression $\text{Tr}[O_{\text{root}} \rho]$ and rather straightforwardly get:

$$\text{Tr}[O_{\text{root}} \rho] = \text{Tr} \left[ \left( V_{\text{root}} \otimes V_{\text{root}}^* \right) \times \rho \right] \tag{39}$$

$$= V_{\text{root}}^* \times \rho \times V_{\text{root}}, \tag{40}$$

which is indeed the same probability as defined in Definition 4.1. $\square$

## C.3 PROOF OF PROPOSITION 4.7

**Proposition 4.7.** *Every entry of a vector $P_k(\mathbf{x}_k)$ forms a valid probability distribution.*

*Proof.* If $P_{root}(\mathbf{x})$ is the computation unit at the root of the layered PC, each entry $i$ of $P_{root}(\mathbf{x})$ forms a probability distribution if $\forall i : P_{root,i}(\mathbf{x}) \geq 0$, for every $\mathbf{x} \in \Omega(\mathbf{X})$ and if $\forall i : \sum_{\mathbf{x} \in \Omega(\mathbf{X})} P_{root,i}(\mathbf{x}) = 1$. The first condition is trivially satisfied as the circuit only performs linear operations on matrices and vectors ($P_{x_k}, W_k$) with positive entries only. For the condition $\sum_{\mathbf{x} \in \Omega(\mathbf{X})} P_{root}(\mathbf{x}) = 1$ we observe that we can simply push down the summation for each variable to the respective leaf unit. This yields the following summation in the leaves:

$$\sum_{x_k \in \Omega(X_k)} W_k \times P_{x_k} = W_k \times \sum_{x_k \in \Omega(X_k)} P_{x_k} = W_k \times \eta_k = \eta_k. \tag{41}$$

Here $\eta$ denotes a vector having as entries only ones (with appropriate dimensions). For the last step in the equation above we exploited the fact that the weight matrices $W_k$ are row-normalized.

Passing on the marginalized leaves to the parent units gives us:

$$W_k \times \left( \eta_{k_l} \otimes \eta_{k_r} \right) = W_k \times \eta_k = \eta_k \tag{42}$$

Repeating this process until we reach the root will eventually result in $\sum_{\mathbf{x} \in \Omega(\mathbf{X})} P_{root}(\mathbf{x}) = \eta_{root}$. This means that all the entries of all the vectors $p_k(\mathbf{x}_k)$ equal to 1 when marginalized, which means in turn that the second condition is satisfied. $\square$

## C.4 PROOF OF PROPOSITION 4.9

**Proposition 4.9.** *All operators $O_k(\mathbf{x}_k)$ in a diagonal PUnC can be represented as diagonal matrices.*

*Proof.* The proof is relatively straightforward as in the leaves we already have diagonal matrices by definition. For an internal unit $k$ we have that

$$O_k = \sum_j J_j D_{kj} \left(O_{kl} \otimes O_{kr}\right) D_{kj}^* J_j^* kj, \tag{43}$$

which can be written as

$$O_k = \sum_j J_j D_{kj} J_j^* kj, \tag{44}$$

where $D_{kj}$ is diagonal and PSD. Carrying out the remaining matrix products then gives us:

$$O_k = \sum_j H_j \operatorname{Tr}[D_{kj}], \tag{45}$$

which is clearly a diagonal matrix (and also PSD). □

## C.5 PROOF OF PROPOSITION 4.10

**Proposition 4.10.** *Probabilistic circuits and diagonal PUnCs are isomorphic.*

*Proof.* In order to show this we need to show that each computation unit in a probabilistic circuit can be mapped to a computation unit in a diagonal PUnC– and vice versa. Starting at the leaves for both circuits we have

$$P_{x_k} \tag{46}$$

for the probabilistic circuit and

$$\Delta_{x_k} \Delta_{x_k}^*, \tag{47}$$

for the diagonal PUnC. Given that $\Delta_{x_k}$ is a diagonal matrix the product $\Delta_{x_k} \Delta_{x_k}^*$ is also a diagonal matrix with positive entries only. Taking into consideration the completeness constraints $\sum_{x_k} P_{x_k} = \eta_k$ ($\eta$ being a vector with only ones) and $\sum_{x_k} \Delta_{x_k} \Delta_{x_k}^* = \mathbb{1}$ is trivial to see that in the leaves probabilistic circuits can be mapped onto diagonal PUnCs and vice versa.

For the internal computation units we need to show that

$$P_k = W_k \times \left(P_{k_l} \otimes P_{k_l}\right) \tag{48}$$

$$\Leftrightarrow P_{ki} = \sum_j W_{kij} \times \left(P_{k_l} \otimes P_{k_l}\right)_j \tag{49}$$

is equivalent to

$$O_k = \Phi(O_{k_l} \otimes O_{k_r}) = \sum_j J_j D_{kj}(O_{k_l} \otimes O_{k_r}) D_{kj}^* J_j^*. \tag{50}$$

We first note that for diagonal PUnCs all the operators $O_k$ are representable as diagonal matrices. This allows us to rewrite Equation 50 as

$$O_k = \Phi(O_{k_l} \otimes O_{k_r}) = \sum_j J_j \underbrace{D_{kj} D_{kj}^*}_{=:\Lambda_{kj}}(O_{k_l} \otimes O_{k_r}) J_j^*. \tag{51}$$

When writing out this sum of matrix product on the right-hand side explicitly it is trivial to observe that the diagonal element $O_{kjj}$ can be written as:

$$O_{kjj} = \operatorname{Tr}\left[\Lambda_{kj} \times (O_{k_l} \otimes O_{k_r})\right], \tag{52}$$

with all the off-diagonal elements being zero. This can in turn be written as

$$O_{kjj} = \sum_i \text{diagvec}(\Lambda_{kj})_i \Big( \text{diagvec}(O_{k_l}) \otimes \text{diagvec}(O_{k_r}) \Big)_i, \tag{53}$$

where $\text{diagvec}(\Lambda_{kj})$ denotes a vector whose entries corresponds to the diagonal elements of $\Lambda_{kj}$.

Switching around the names of the indices $i$ and $j$ we can write:

$$\text{diagvec}(O_k)_i = \sum_j \text{diagvec}(\Lambda_{ki})_j \Big( \text{diagvec}(O_{k_l}) \otimes \text{diagvec}(O_{k_r}) \Big)_j. \tag{54}$$

We now identify the expressions in Equation 49 and Equation 54 as follows with each other:

$$\text{diagvec}(O_k)_i = P_{ki} \tag{55}$$
$$\text{diagvec}(\Lambda_{ki})_j = W_{kij} \tag{56}$$
$$\text{diagvec}(O_{k_l})_i = P_{k_l i} \tag{57}$$
$$\text{diagvec}(O_{k_r})_i = P_{k_r i}. \tag{58}$$

This – together with the equivalence between $\forall k : \text{Tr}[D_{ki}D_{ki}^*] = \text{Tr}[\Lambda_{ki}] = 1$ and the row-normalization of the $W_k$'s – establishes the isomorphism between the two circuit classes. $\square$

# D  NOISY AND BLOCK-DIAGONAL PUNCS

In order to increase the expressive power of non-monotone circuits, Loconte et al. [2025b] proposed to multiply a pure PUnC with a probabilistic circuit. This was motivated by their observation that the expressive power of monotone and squared circuits (a special case of pure PUnCs) are incomparable [de Colnet and Mengel, 2021]. That is, each of these circuit classes can express circuits that would lead to an exponential blow-up in the respective other circuit class. The idea is relatively simple: they express an unnormalized probability distribution as the product of a probabilistic circuit and a traced PUnC:

$$q(\mathbf{x}) \text{Tr}[O(\mathbf{x})\rho]. \tag{59}$$

They called such a model a $\mu$SOCS. We will put these models in a quantum information theoretic framework. This will then lead to the realization that $\mu$SOCSs are nothing but block-diagonal operator circuits.

## D.1  SUB-COMPLETE PROBABILITY DISTRIBUTIONS

An important concept in quantum information theory that generalizes the standard POVM (cf. Definition 2.2) is the so-called noisy POVM, which enables for instance modelling imperfect measurement devices. In the context of machine learning this might be an improperly labeled data point of an error in the detector, e.g. a pixel flip in the camera that took a picture.

**Definition D.1.** A noisy positive operator-valued measure is a set of PSD matrices $\{E(i)\}_{i=0}^{I-1}$ ($I$ being the number of possible measurement outcomes) such that:

$$\sum_{i=0}^{I-1} E(i) = M < \mathbb{1}, \tag{60}$$

where the inequality sign is interpreted using the Loewner ordering of PSD matrices, i.e. $M < \mathbb{1} \Leftrightarrow \mathbb{1} - M$ is PSD.

**Definition D.2** (Sub-complete Probability Distribution). Let $\mathbf{X}$ be a set of random variables with sample space $\Omega(\mathbf{X})$. We call a probability distribution *sub-complete* if $\forall \mathbf{x} \in \Omega(\mathbf{X}) : p(\mathbf{x}) \geq 0$ and if $\sum_{\mathbf{x} \in \Omega(\mathbf{X})} p(\mathbf{x}) \leq 1$. We call a probability distribution *strictly sub-complete* if the latter inequality holds strictly and *complete* if equality holds.

**Proposition D.3.** *Noisy POVMs induce a sub-complete probability distributions.*

*Proof.* Let $\rho$ be a density matrix and $\{E(i)\}_{i=0}^{I-1}$ be a POVM such that $\sum_{i=0}^{I-1} E(i) = M$. We then have

$$\text{Tr}[(\mathbb{1} - M)\rho] \geq 0, \tag{61}$$

holds as $\mathbb{1} - M$ is PSD by definition. Pushing the trace over the subtraction and rearranging terms yields:

$$\text{Tr}[\mathbb{1}\rho] - \text{Tr}[M\rho] \geq 0 \Leftrightarrow \text{Tr}[M\rho] \leq 1. \tag{62}$$

This means that the induced probability distribution $p(i) = \text{Tr}[E(i)\rho]$ is indeed sub-complete. $\square$

To construct sub-complete probability distributions using operator circuits we now introduce *noisy positive unital circuits* or NoisePUnCs.

**Definition D.4.** A NoisePUnC $Q(\mathbf{x})$ is of the form

$$Q(\mathbf{x}) = q(\mathbf{x})O(\mathbf{x}), \tag{63}$$

where $O(\mathbf{x})$ is a PUnC and $q(\mathbf{x})$ is real-valued and belongs to the $[0, 1]$ interval for every $\mathbf{x}$.

**Proposition D.5.** *NoisePUnCs induce sub-complete probability distributions.*

*Proof.* Let $\mathbf{X}$ be a set of random variables and $Q(\mathbf{x}) = q(\mathbf{x})O(\mathbf{x})$ with $\mathbf{x} \in \Omega(\mathbf{X})$ be a NoisePUnC inducing a probability distribution

$$\pi_{\mathbf{X}}(\mathbf{x}) = \mathrm{Tr}[q(\mathbf{x})O(\mathbf{x})\rho]. \tag{64}$$

As $q(\mathbf{x})$ is a scalar we can write

$$\pi_{\mathbf{X}}(\mathbf{x}) = q(\mathbf{x}) \mathrm{Tr}[O(\mathbf{x})\rho] = q(\mathbf{x})p_X(\mathbf{x}), \tag{65}$$

where $p_X(\mathbf{x})$ is the (complete) probability distribution induced by the set $\{O(\mathbf{x})\}_{\mathbf{x} \in \Omega(\mathbf{X})}$. As $0 \leq q(\mathbf{x}) \leq 1$ (by definition) and $0 \leq p_{\mathbf{X}}(\mathbf{x}) \leq 1$ we have also that $0 \leq \pi_{\mathbf{X}}(\mathbf{x}) \leq 1$. This shows that each event $\mathbf{x} \in \Omega(\mathbf{X})$ has a positive probability.

For the sub-completeness we observe that the sum of positive terms

$$\sum_{\mathbf{x} \in \Omega(\mathbf{X})} p_{\mathbf{X}}(\mathbf{x}) = 1. \tag{66}$$

If we weigh each term in the sum with a scalar $q(\mathbf{x}) \in [0, 1]$ we immediately conclude that $\sum_{\mathbf{x} \in \Omega(\mathbf{X})} \pi_{\mathbf{X}}(\mathbf{x}) \leq$ and that we have indeed a sub-complete probability distribution. $\square$

## D.2 NOISE VARIABLES

All of this is a bit unnerving. How is it possible that we have probabilistic events $\mathbf{x} \in \Omega(\mathbf{X})$ that violate the second of Kolmogorov's probability axioms (by having sub-complete distributions)? The resolution to this problem lies in interpreting sub-complete distributions as joint distributions not only over the set of variables $\mathbf{X}$ but an extra set of variables $\mathbf{Y}$ representing extra noise [Wiseman and Milburn, 2009]

$$\pi_{\mathbf{X}}(\mathbf{x}) = p_{\mathbf{X}\mathbf{Y}}(\mathbf{x}, \mathbf{y}), \tag{67}$$

for which we indeed have

$$\sum_{\mathbf{x} \in \Omega(\mathbf{X})} \sum_{\mathbf{y} \in \Omega(\mathbf{Y})} p_{\mathbf{X}\mathbf{Y}}(\mathbf{x}, \mathbf{y}) = 1. \tag{68}$$

The problem with the random variables $\mathbf{Y}$ is that they are not accessible to the observer. In the sense that they are observable in principle but in practice not, e.g. a defect in a sensor.[1] However, the probability we are actually interested in is the one for $\mathbf{X}$ given $\mathbf{Y}$

$$
\begin{aligned}
p_{\mathbf{X}|\mathbf{Y}}(\mathbf{x} \mid \mathbf{y}) &= \frac{p_{\mathbf{X}\mathbf{Y}}(\mathbf{x}, \mathbf{y})}{\sum_{\mathbf{x} \in \Omega(\mathbf{X})} p_{\mathbf{X}\mathbf{Y}}(\mathbf{x}, \mathbf{y})} \\
&= \frac{\pi_{\mathbf{X}}(\mathbf{x})}{\sum_{\mathbf{x} \in \Omega(\mathbf{X})} \pi_{\mathbf{X}}(\mathbf{x})}
\end{aligned}
\tag{69}
$$

The question now became whether we can choose $q(\mathbf{x})$ such that the summation in the denominator is tractable, i.e. can be performed in time polynomial in the number of random variables $\mathbf{X}$.

---

[1] Note these practically unobservable variables should not be confused with the concept of (local) hidden variables, which lead to violations of Bell's inequality.

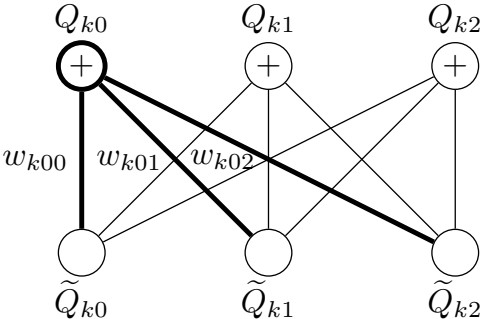

Figure 4: Graphical representation of operator mixing as described in Equation 72. The operators $Q_{k0}$, $Q_{k1}$, and $Q_{k2}$ are all PSD matrices and constructed using the operators $\widetilde{Q}_{k0}$, $\widetilde{Q}_{k1}$, and $\widetilde{Q}_{k2}$ using weighted sums. For $Q_{k0}$ we also indicate the mixing weights $w_{k00}$, $w_{k01}$, and $w_{k02}$, which are positive real-valued scalars and satisfy $w_{k00} + w_{k01} + w_{k02} = 1$.

In the probabilistic circuit literature it has been shown that the product of two circuits is representable as a single circuit within polytime and polyspace if the two circuits are compatible [Khosravi et al., 2019, Vergari et al., 2021]. Importantly, this single circuit then allows for summing out the variables $\mathbf{x}$. As pointed out also by Loconte et al. [2025b], all we need is that $q(\mathbf{x})$ and $O(\mathbf{x})$ are both structured decomposable according to the same partition tree (or variable tree). Furthermore, we pick $q(\mathbf{x})$ to be a probabilistic circuit and $O(\mathbf{x})$ to be a PUnC. For the sake of simplicity we construct the circuit $q(\mathbf{x})$ (with an underlying partition circuit) using Hadamard products instead of Kronecker products.

$$
q_k(\mathbf{x}_k) = \begin{cases} f_{x_k}, & \text{if } k \text{ leaf} \\ W_k \times \left( q_{k_l}(\mathbf{x}_{k_l}) \odot q_{k_r}(\mathbf{x}_{k_r}) \right) & \text{else.} \end{cases}
$$

### D.3 NOISEPUNCS AS DEEP OPERATOR MIXTURES

An interesting consequence of modelling noise in the system via a probabilistic circuit $q(\mathbf{x})$ is that it effectively results in a model that can be interpreted as a recursive mixture of positive operators. To see this, let us first write the matrix vector product in the computation units of $q(\mathbf{x})$ using explicit indices:

$$
q_k = W_k \times (q_{k_l} \odot q_{k_r}) \Leftrightarrow q_{ki} = \sum_j w_{kij} q_{k_l j} q_{k_r j}. \tag{70}
$$

Here $q_{ki}$ denotes the $i$-th entry of the vector $q_k$ and $w_{kij}$ denotes the elements of the matrix $W_k$. We also dropped the explicit dependency on $\mathbf{x}_k$.

We now multiply each $q_{ki} \in [0, 1]$ with the corresponding operator $O_k$

$$
\begin{aligned}
q_{ki} O_k &= \left( \sum_j w_{kij} q_{k_l j} q_{k_r j} \right) \Phi_k(O_{k_l} \otimes O_{k_r}) \\
&= \sum_j w_{kij} \underbrace{\Phi_k(q_{k_l j} O_{k_l} \otimes q_{k_r j} O_{k_r})}_{= \widetilde{Q}_{kj}}
\end{aligned} \tag{71}
$$

This means that we have for each computation unit $k$ not one but multiple operators, indexed by $i$, and each of the $i$ operators is a mixture of operators:

$$
Q_{ki}(\mathbf{x}_k) = \sum_j w_{kij} \widetilde{Q}_{kj}(\mathbf{x}_k). \tag{72}
$$

We illustrate the situation in Figure 4. Such mixtures of operators can be regarded as generalizing standard mixture models of probability distributions, which can be recovered by considering the special case of operators of dimension one – in other words positive real-valued scalars. With this interpretation we can follow a similar reasoning to viewing probabilistic circuits as PUnCs closed over diagonal matrices. But now we view NoisePUnCs as PUnCs closed over block-diagonal matrices.

# E   PROOFS FOR SECTION 5

## E.1   PROOF FOR THEOREM 5.8

**Theorem 5.8.** *Let* $\mathbf{X}_k$ *be a set of random variables with sample space* $\Omega(\mathbf{X}_k)$ *equal to the possible values for* $\mathbf{x}_k$. *A D-PUnC* $o(\mathbf{x}_k)$ *and a density matrix* $\rho$ *then define a proper probability distribution as* $p_{\mathbf{X}}(\mathbf{x}) = \mathrm{Tr}[o(\mathbf{x})\rho]$.

*Proof.* The proof is relatively straightforward as positivity is guaranteed by the fact that we perform computations on members of POVMs that all preserve positive semi-deifiniteness (Kronecker product in the product units and convex combination of quantum operations in the sum units). This guarantees that $o(\mathbf{x})$ is PSD and hence $\mathrm{Tr}[o(\mathbf{x})\rho] > 0$. The argument is similar for the completeness of $\mathrm{Tr}[o(\mathbf{x})\rho]$ as here we need to show that $\sum_{\mathbf{x} \in \Omega(\mathbf{X})} o(\mathbf{x}) = \mathbb{1}$. For this we push down the summations to the corresponding leaves where we obtain unit matrices. The circuit then performs again computation with these unit matrices that are all unital, which guarantees indeed that

$$\sum_{\mathbf{x} \in \Omega(\mathbf{X})} \mathrm{Tr}[o(\mathbf{x})\rho] = \mathrm{Tr}\left[\left(\sum_{\mathbf{x} \in \Omega(\mathbf{X})} o(\mathbf{x})\right)\rho\right] \tag{73}$$
$$= \mathrm{Tr}\left[\mathbb{1}\rho\right]$$
$$= 1$$

$\square$

## E.2   PROOF OF PROPOSITION 5.9

**Proposition 5.9.** *SD-PUnCs are a proper subset of D-PUnCs.*

*Proof.* Consider the expression of a sum unit in a D-PUnC

$$\sum_{j \in \mathrm{in}(k)} w_{kj}\Phi_{kj}(o_j(\mathbf{x}_j)). \tag{74}$$

Writing also the operation that is performed in the product units we obtain:

$$\sum_{j \in \mathrm{in}(k)} w_{kj}\Phi_{kj}\left(o_{j_l}(\mathbf{x}_{j_l}) \otimes o_{j_r}(\mathbf{x}_{j_r})\right). \tag{75}$$

Given that all product units decompose in the same way we can write this as

$$\sum_{j \in \mathrm{in}(k)} w_{kj}\Phi_{kj}\left(o_{j_l}(\mathbf{x}_{k_l}) \otimes o_{j_r}(\mathbf{x}_{k_r})\right), \tag{76}$$

where the index on $\mathbf{x}_j$, $\mathbf{x}_{j_l}$, and $\mathbf{x}_{j_r}$ is now on $k$ and not on $j$. Given that the expression above is a completely positive map [Nielsen and Chuang, 2001] that maps positive semi-definite matrices to positive semi-definite matrices, we can write the expression using a single quantum operation:

$$\Phi_k\left(o_{k_l}(\mathbf{x}_{k_l}) \otimes o_{k_r}(\mathbf{x}_{k_r})\right). \tag{77}$$

Following the convention from Section 3 and using upper case letter for the circuit we have:

$$\Phi_k\left(O_{k_l}(\mathbf{x}_{k_l}) \otimes O_{k_r}(\mathbf{x}_{k_r})\right), \tag{78}$$

which corresponds exactly to the computations performed in an SD-PUnC and thereby concludes the proof. $\square$