# OpenReview forum: "A Quantum Information Theoretic Approach to Tractable Probabilistic Models"
_auai.org/UAI/2025/Workshop/TPM — TPM 2025_

### Official Review · Reviewer_HZap · 2025-06-10
**Perfect Fit For TPM Workshop**

**Rating:** 3

**Review:**

This paper introduces Positive Unital Circuits (PUnCs), a novel class of tractable probabilistic models grounded in quantum information theory that generalize and unify existing circuit classes like probabilistic and PSD circuits. PUnCs support efficient inference, encode logical constraints, and expand the expressive power of tractable models by relaxing structured decomposability. These contributions align directly with the TPM workshop’s goals of scaling reliable probabilistic reasoning and bridging logical and probabilistic AI. The acceptance of this work at the main UAI conference signifies that its validity and quality have already been rigorously assessed and acknowledged as outstanding.

---

### Official Review · Reviewer_eC7S · 2025-06-17
**A Quantum Information Theoretic Approach to Tractable Probabilistic Models**

**Rating:** 3

**Review:**

This paper establishes a connection between probabilistic circuits and quantum information theory, and using this connection introduces positive unital circuits which generalize existing monotone and non-monotone circuits. The topic is clearly relevant to the TPM community. I think a table/figure to visually summarize PUnC classes and their relation to existing circuit classes would be very helpful. I also found it a bit odd that PCs and their properties are formally defined only in Section 5 while a special case of structured decomposable PCs was already discussed before.